# SELF-SUPERVISED OFF-POLICY RANKING VIA CROWD LAYER

## ABSTRACT

Off-policy evaluation (OPE) aims to estimate the online performance of target policies given dataset collected by some behavioral policies. OPE is crucial in many applications where online policy evaluation is expensive. However, existing OPE methods are far from reliable. Fortunately, in many real-world scenarios, we care only about the ranking of the evaluating policies, rather than their exact online performance. Existing works on off-policy ranking (OPR) adopt a supervised training paradigm, which assumes that there are plenty of deployed policies and the labels of their performance are available. However, this assumption does not apply to most OPE scenarios because collecting such training data might be highly expensive. In this paper, we propose a novel OPR framework called SOCCER, where the existing OPE methods are modeled as workers in a crowdsourcing system. SOCCER can be trained in a self-supervised way as it does not require any ground-truth labels of policies. Moreover, in order to capture the relative discrepancies between policies, we propose a novel transformer-based architecture to learn effective pairwise policy representations. Experimental results show that SOCCER achieves significantly high accuracy in a variety of OPR tasks. Surprisingly, SOCCER even performs better than baselines trained in a supervised way using additional labeled data, which further demonstrates the superiority of SOCCER in OPR tasks.

## 1 INTRODUCTION

Off-policy evaluation (OPE) aims to estimate online performance of given policies using only historical data collected by some other behavior policies. It is crucial to deploying reinforcement learning (RL) to real-world applications, such as trading, advertising, autonomous vehicles and drug trials, where online policy evaluation might be highly expensive. OPE also becomes increasingly important in causal inference and model selection for offline RL (Oberst & Sontag, 2019; Nie et al., 2021).

Most existing works on OPE focus on estimating the online performance of target policies and can be categorized into three classes: Inverse Propensity Scoring (IPS) based methods, Direct Methods (DM) and Hybrid Methods (HM). Unfortunately, existing OPE methods are far from reliable in real applications. Standard IPS based estimators such as importance sampling suffer from high variance due to the product of importance weights (Hanna et al., 2019). DM requires extra estimators of environmental dynamics or value functions, which are hard to learn when the observation data is high-dimensional or insufficient. HM such as doubly robust estimators combine IPS and DM (Jiang & Li, 2016), yet it often comes with additional hyperparameters that need to be carefully chosen.

Fortunately, in many real-world scenarios, we do not need to estimate the exact online performance of target policies. Instead, we only care about which policy would perform the best when deployed online. This inspires us to develop a policy ranker that focuses on predicting the ranking of target policies regarding to their online performance. A recent work proposes a policy ranking model called SOPR-T (Jin et al., 2022), which is trained in a supervised paradigm under the assumption that there are plenty of extra deployed policies whose performance can be used as supervision signals. However, this assumption is impracticable in many real-world OPE tasks since collecting online performance of policies can be extremely expensive. In addition, SOPR-T directly maps the data of state-action pairs to a score, yielding a low-efficient policy representation scheme which fails to capture the relative discrepancies between policies.

In this paper, we propose a novel Self-supervised Off-poliCy ranking model based on Crowd layER (SOCCER) to address the above challenges. The novelty of SOCCER is two-fold. First, we employ a crowdsourcing paradigm to solve the OPR problem, where the workers come from a diverse pool of existing OPE methods, who provide labels of whether a policy would perform better than another one. Note that these labels are constructed by comparing the estimated accumulated rewards of the target policies, thus our model can be trained in a self-supervised way. Moreover, we propose a novel Policy Comparison Transformer (PCT) architecture to learn efficient policy representations. Instead of directly mapping the state-action pairs to a policy embedding (as is done in SOPR-T), PCT learns pairwise representation of two policies capturing difference of them at the same set of states. With the help of PCT, our policy ranking model generalizes well in the policy space. Experimental results show that SOCCER not only achieves significant higher ranking performance than existing OPE methods, but also outperforms baselines trained using additional ground-truth labels.

## 2 RELATED WORKS

**Off-policy evaluation/ranking.** The goal of OPE is to precisely predict the online performance of target policies given trajectory data collected by some other behavior policies. Standard importance sampling approach suffers from exponential variance with respect to the time horizon (Li et al., 2015; Jiang & Li, 2016). Recent works such as Fitted-Q evaluation (Hoang et al., 2019) and marginalized importance sampling (Liu et al., 2018) achieve polynomial variance, yet they rely on additional function approximators. Direct methods avoid the large variance by learning the dynamic model or Q-function, which could be biased especially when the data is insufficient. Some works study the hyperparameter-free policy selection problem, yet their method only applies to Q-learning based policies (Zhang & Jiang, 2021). A recent work directly studies the OPR problem, where it collects online performance of a large set of policies and uses these labeled data to train a policy ranker (Jin et al., 2022). However, collecting such data might be extremely expensive in many applications.

**Learning from crowds.** Crowdsourcing systems enable machine learners to collect labels of large datasets from crowds. One big issue with crowdsourcing systems is that the labels provided by crowds are often noisy (S. & Zhang, 2019). To tackle this challenge, various probabilistic generative methods are proposed for statistical inference (Yuchen et al., 2016; Tian & Zhu, 2015). Another line of works use discriminative models that find the most likely label for each instance (Jing et al., 2014; 2015). A recently work called Crowd Layer (CL) first describes an algorithm for jointly learning the target model and the reliability of workers (Filipe & Pereira, 2018). CL proposes a simple yet efficient crowd layer that can train deep neural networks end-to-end directly from the noisy labels. In our work, we treat existing OPE methods as workers and adopt CL to process multiple noisy labels, because CL is naturally compatible with our model.

**Policy representation.** Compact but informative representations of policies not only benefit the policy learning process (Tang et al., 2022), but also help with the policy transfer among different tasks (Isac et al., 2019; G. et al., 2017). A straightforward idea is to represent a policy by its network parameters, yet this leads to a very sparse representation space. Network Fingerprint (Harb et al., 2020) proposes a differentiable representation that uses the concatenation of the vectors of actions outputted by the policy network on a set of probing states. Some recent works try to encode policy parameters as well as state-action pair data into a low-dimensional embedding space (Tang et al., 2022; Jin et al., 2022). However, existing works focus on single policy representations, which fail to capture the relative discrepancies between policies.

## 3 PROBLEM STATEMENT

**Markov decision process.** We consider the underlying environment as a Markov decision process (MDP) and define an MDP as a finite–horizon tuple $\mathcal{M} = (\mathcal{S}, \mathcal{A}, \mathcal{T}, \mathcal{P}, \mathcal{R}, \gamma)$. Here, $\mathcal{S}$ is the state space, and $\mathcal{A}$ is the action space. $\mathcal{T}$ is the length of time horizon. $\mathcal{P}$ and $\mathcal{R}$ are the transition function and the reward function, respectively. $\mathcal{P}(s_{t+1}|s_t, a_t)$ represents the probability of transitioning from state $s_t$ to state $s_{t+1} \in \mathcal{S}$ when the agent takes action $a_t \in \mathcal{A}$ under state $s_t \in \mathcal{S}$ and $\mathcal{R}(s_t, a_t)$ represents the immediate reward the agent receives. The expected return of a policy $\pi$ can be computed by $\mathbb{E}_{\mathcal{P}}[\sum_{t=1}^{\mathcal{T}}[\gamma^t \mathcal{R}(s_t, \pi(s_t))]]$, where $\gamma \in (0, 1]$ is the discount factor.

**Off-policy ranking.** The goal of OPE is to estimate the expected return of a policy $\pi$ without deploying it online, given an offline dataset $\mathcal{D} = \{\tau_i\}_{i=1}^N$, where $\tau_i = (s_{i,0}, a_{i,0}, r_{i,0}, \cdots, s_{i,\mathcal{T}}, a_{i,\mathcal{T}}, r_{i,\mathcal{T}})$ are trajectories generated by some behavior policies. OPE is usually used for model selection: We are required to select the most promising policy from a candidate set of available policies before actual deployment. Take recommender systems as example, we can easily obtain a set of candidate policies by adjusting the training data or the hyperparameters of the model. However, we often need to select very few policies from the candidates for online test, since a bad policy would harm the user experience. Therefore, we care more about the ranking of the candidate policies, instead of their exact expected reward. We formally define the off-policy ranking problem as follows.

**Definition 3.1** (Off-Policy Ranking, OPR). Given a set of trajectory data $\mathcal{D} = \{\tau_i\}_{i=1}^N$ generated by some behavior policies and a set of target policies $\Pi = \{\pi_j\}_{j=1}^M$, an OPR algorithm outputs a ranking of the target policies that aligns with their online expected accumulated rewards.

Intuitively, OPR should be easier than OPE, since the solution of OPE also implies the solution of OPR. However, OPR faces some unique challenges. First, since the policy space might be extremely large, we need efficient policy representations that capture their relative differences so that the policy ranker could generalize across the policy space. Second, we lack the ground-truth ranking of the policies in the training set, thus the direct supervised learning approaches do not apply. We will elaborate on how we address these challenges in Section 4.

## 4 Approach

In this section, we elaborate on how the OPR problem can be addressed under our SOCCER framework. SOCCER takes the offline trajectory data $\mathcal{D}$ and two target policies $\pi_i$, $\pi_j$ as inputs and outputs the probability of whether $\pi_i$ would perform better than $\pi_j$. We begin by introducing how to learn effective pairwise policy representations under a novel transformer-based architecture. Then, we will introduce how to train our model using the labels provided by existing OPE methods.

### 4.1 Learning pairwise policy representations

**Pairwise Policy Representation.** A policy is generally considered as a conditional distribution over actions given current state. Therefore, a policy can be naturally represented by a set of state-action pairs where the actions are sampled from the policy. However, such a straightforward policy representation could be inefficient since the number of state-action pairs can be extremely large. Previous works address this issue by extracting high-level features from the state-action pairs using deep neural networks (Jin et al., 2022). Although these representations reflect the features of single policies, they fail to capture the discrepancies of different policies at some crucial states.

To this end, we aim to learn pairwise policy representations by comparing two policies' decisions at the same set of states. Formally, given a set of states $\{s_1, ..., s_K\}$ and two policies $\pi_i$, $\pi_j$, we can construct the following sequence of state-action pairs by taking actions at these states:

$$\tau_{i,j} = <(s_1, a_1^i), (s_1, a_1^j), \cdots, (s_K, a_K^i), (s_K, a_K^j)>, \tag{1}$$

where $a^i \sim \pi_i(\cdot|s)$, and $a_j \sim \pi_j(\cdot|s)$. We denote by $\chi_{i,j} = g(\tau_{i,j}) \in \mathbb{R}^n$ the pairwise policy representation where $g$ is a function that maps $\tau_{i,j}$ to an $n$-dimensional representation space. Since our goal is to predict whether $\pi_i$ performs better than $\pi_j$, therefore a pairwise policy representation should indicate the order of the two policies. We regard $\chi_{i,j}$ and $\chi_{j,i}$ as different representations and will show how to learn them using transformers. In addition, since the datasets are often very large, computing the policy representations using all the states can be extremely slow. In practice, we use a sampled set of states to compute the approximated representations during training, and take the averaged output results of multiple samples as the final representation during inference.

**Policy Comparison Transformer (PCT).** Transformers are proved to be effective for learning dependencies between different positions in sequences. Prior works has employed transformers to extract features from trajectory sequences (Lili et al., 2021; Michael et al., 2021). However, existing transformer architectures fail to capture the differences of two policies' decisions. In our work, we propose the PCT architecture to learn pairwise policy representations. Unlike previous works where

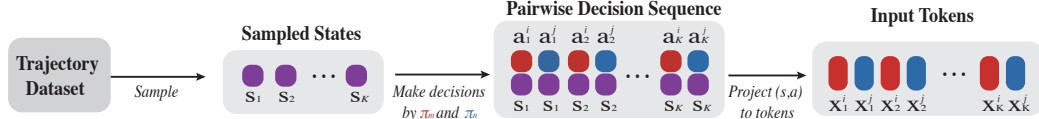

Figure 1: **Construction of input tokens.**

the positional encodings indicate the positions of state-action pairs in a trajectory, PCT uses positional encoding to distinguish the relative order of two policies. In this way, the learned pairwise policy representation $\chi_{i,j}$ can be directly used to predict whether $\pi_i$ performs better than $\pi_j$.

Figure 1 shows the construction of input tokens. We first sample $K$ states from $\mathcal{D}$ and then use a linear encoder $f$ to map the $K$ state-action pairs into $2K$ tokens:

$$\mathbf{x}_k^i = f(s_k, a_k^i), \quad \mathbf{x}_k^j = f(s_k, a_k^j), \quad k = 1, ..., K \tag{2}$$

where $i$ and $j$ represent the indexes of two policies. In order to represent the relative order of $\pi_i$ and $\pi_j$, we introduce two one-hot positional encodings $\mathbf{e}_\alpha = [1, 0]$ and $\mathbf{e}_\beta = [0, 1]$, where $\mathbf{e}_\alpha$ indicates the policy ranked higher and $\mathbf{e}_\beta$ indicates the policy ranked lower. We also use an aggregation token $\mathbf{e}_0$, which is a learnable vector for aggregating the information from the other $2K$ tokens (Zhu et al., 2021). The final inputs that indicate $\pi_i$ ranked higher than $\pi_j$ can be represented as:

$$\mathbf{z_{i>j}} = [\mathbf{e}_0, \mathbf{x}_1^i + \mathbf{e}_\alpha, \mathbf{x}_1^j + \mathbf{e}_\beta, \cdots \mathbf{x}_K^i + \mathbf{e}_\alpha, \mathbf{x}_K^j + \mathbf{e}_\beta] \tag{3}$$

This construction of inputs has two advantages. First, the two policies share the same set of states, thus their discrepancies are naturally represented by the different actions taken at these states. Second, we can easily get a mirrored representation $z_{j>i}$ by simply exchange the positional encoding $\mathbf{e}_\alpha$ and $\mathbf{e}_\beta$ used in $z_{j>i}$ .

We adopt a widely used transformer architecture as our encoder (Dosovitskiy et al., 2021). It contains $L$ alternating layers of multi-head self-attention (MSA) and multi-layer perception (MLP) blocks. Layernorm (LN) and residual connections are applied to the outputs of each block. For brevity, we re-write the inputs in Equation 3 as $\mathbf{z}^{(0)}$. And the computations at each block can be represented as:

$$\hat{\mathbf{z}}^0 = \text{MSA}(\text{LN}(\mathbf{z}^{(l-1)})) + \mathbf{z}^{(l-1)} \qquad l = 1, \cdots, L$$
$$\mathbf{z}^l = \text{MLP}(\text{LN}(\hat{\mathbf{z}}^{(l-1)})) + \hat{\mathbf{z}}^{l-1} \qquad l = 1, \cdots, L \tag{4}$$
$$\chi_{i,j} = \text{LN}(\mathbf{z}_L^0).$$

The final pairwise policy representation $\chi_{i,j}$ is the corresponding outputs of the aggregation token $\mathbf{e}_0$ taken from $\mathbf{z}^{(L)}$. Note that $\chi_{i,j}$ changes to $\chi_{j,i}$ when we exchange the positional encodings $\mathbf{e}_\alpha$ and $\mathbf{e}_\beta$, but they are permutation invariant to the order of inputted state-action pairs.

## 4.2 A CROWDSOURCING APPROACH TO OPR

In this section, we will introduce how to train the PCT in two cases regarding to the existence of ground-truth ranking labels of policies. First, we show that the policy ranking problem can be reduced to a binary classification problem since our pairwise policy representations can be directly used to predict the ranking of two policies. Second, we will introduce a novel crowdsourcing system where multiple OPE methods are modeled as workers. We will also show how to train the PCT leveraging the inaccurate labels provided by the workers.

**Reducing OPR to binary classification.** We first consider the case when there is a training set $\Pi' = \{(\pi_i', \mathfrak{R}_i)\}_{i=1}^T$ consisting of $T$ deployed policies $\pi_i'$ as well as their real expected accumulated rewards $\mathfrak{R}_i$. In this case, we can directly construct binary labels by comparing the performance of the two policies. We use an indicator $\mathbb{1}_{\mathfrak{R}_i > \mathfrak{R}_j}$ to represent the label of a pair of policies $(\pi_i, \pi_j)$. The PCT can be trained by minimizing the following binary cross entropy loss:

$$\mathcal{L}_{sup} = - \underset{\pi_i, \pi_j \sim \Pi'}{\mathbb{E}} [(\mathbb{1}_{\mathfrak{R}_i > \mathfrak{R}_j}) \cdot \log(\hat{y}_{i,j}) + (\mathbb{1}_{\mathfrak{R}_i \leq \mathfrak{R}_j}) \cdot \log(1 - \hat{y}_{i,j})], \tag{5}$$

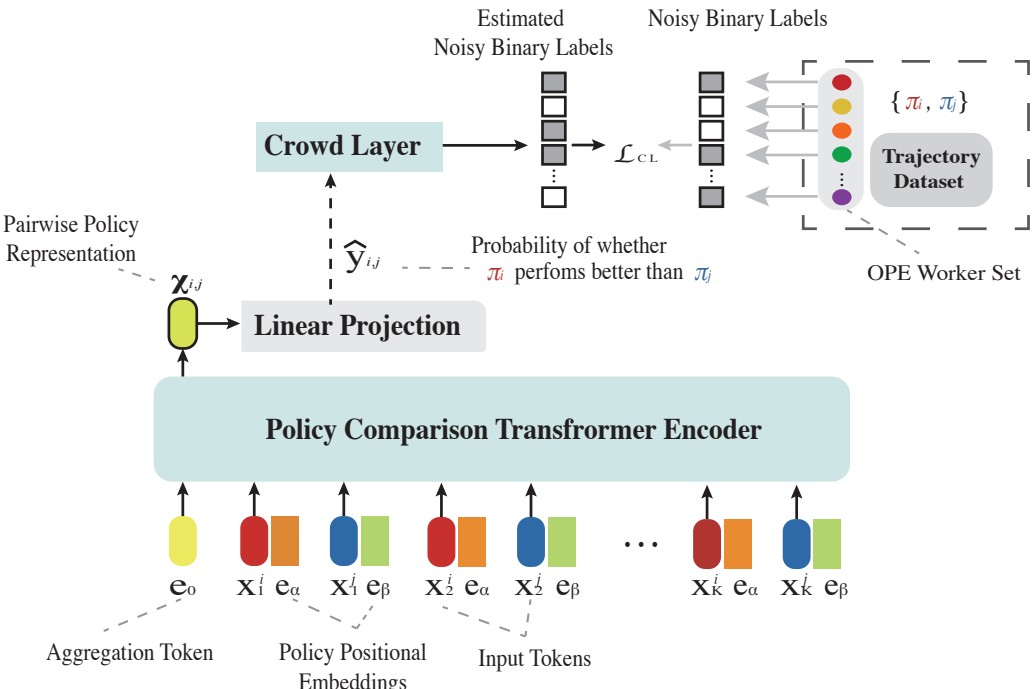

Figure 2: **The framework of SOCCER.**

where $\hat{y}_{i,j} = \texttt{sigmoid}(\phi(\chi_{i,j}))$ represents the predicted probability that $\pi_i$ performs better than $\pi_j$. $\phi$ is a function that projects $\chi_{i,j}$ to a real number. The final ranking of test policies is based on their scores computed by:

$$score_i = \frac{1}{N} \sum_{j \neq i} \hat{y}_{i,j}, i = 1, ..., N, \qquad (6)$$

which can be interpreted as the expected probability that $\pi_i$ performs better than other test policies Rodrigo et al. (2019).

**Learning from crowds by Crowd Layer.** Supervised training is efficient when the dataset $\Pi' = \{(\pi'_i, \mathfrak{R}_i)\}_{i=1}^T$ contains enough policies. Unfortunately, collecting such training data can be extremely expensive in many real applications. Meanwhile, we note that although existing OPE methods are not robust enough, they actually provide candidate solutions to the OPR problem. To this end, we aim to borrow ideas from crowdsourcing domain as an alternative way to approach the OPR problem. Specifically, suppose that there exists a set of OPE algorithms estimating the policy performance, we can treat these algorithms as crowd workers for generating inaccurate labels and make use of these labels to train our models. The intuition is that the inaccurate labels generated by OPE annotators are implicitly conditioned on the ground-truth performance of policies. If we can take advantage of these labels and learn their relationships with the ground-truth labels, our prediction $\hat{y}_{i,j}$ would be more close to the ground-truth labels.

In the framework of SOCCER, we adopt Crowd Layer (CL, (Filipe & Pereira, 2018)) as our backbone for learning from crowd labels. CL is able to automatically distinguish the good from the unreliable annotators and capture their individual biases in many other domains, such as image annotation (Guan et al., 2018; Li et al., 2022) and music genre classification (Rodrigues et al., 2013). In addition, CL is naturally compatible with deep learning approaches since it simply adds a crowd layer to the deep neural networks and can be trained in an end-to-end way. As shown in Figure 2, we add CL to the top of our predicted probability $\hat{y}_{i,j}$. During training, CL adjusts the gradients coming from these noisy labels according to its reliability by scaling them and adjusting their bias. The adjusted gradients are then backpropagated to PCT according to the chain rule.

Formally, assume that there are $R$ annotators of OPE methods. For each annotator $r$, its estimation about the expected return of $\pi_m$ is denoted as $\mathfrak{R}_m^r$. The goal of CL is to train a mapping function $\hat{y}_{i,j}^r = \zeta^r(\hat{y}_{i,j})$ to predict the noisy binary label generated by annotator $r$: $y_{i,j}^r = \mathbb{1}_{\mathfrak{R}_i^r > \mathfrak{R}_j^r}$. The overall objective can be written as:

$$\mathcal{L}_{CL} = - \mathop{\mathbb{E}}_{\substack{r=1,\cdots,R \\ \pi_i,\pi_j \sim \Pi'}} [y_{i,j}^r \cdot \log(\hat{y}_{i,j}^r) + (1 - y_{i,j}^r) \cdot \log(1 - \hat{y}_{i,j}^r).] \tag{7}$$

The complete training procedures of SOCCER is summarized in Algorithm 1. In practice, to reduce the computational cost brought by CL, we set $\zeta^r$ as a linear projection followed by a `sigmoid` function. Therefore, the number of additional parameters only grows linearly with the number of annotators. Note that the CL is only available during training since we still use $\hat{y}_{i,j}$ to generate the predicted ranking of test policies.

---

**Algorithm 1:** SOCCER Training

---

**Data:** Offline Trajectory set $\mathcal{D}$, Policy set $\Pi$, OPE worker set $\mathcal{W}$.
**Result:** Policy comparison transformer $g$ .
$N \leftarrow n$;    `# The number of trainning epochs`
$K \leftarrow k$;    `# The number of sampled states`
Initialize the transformer $g$;
Initialize the aggregation token $\mathbf{e}_0$;
Initialize the policy positional embeddings $\mathbf{e}_\alpha$ and $\mathbf{e}_\beta$;
Initialize the token encoder $f$;
Initialize the linear projection function $\phi$;
Initialize the crowd layer $\xi$;    `# Initialize parameters`
**while** $N \neq 0$ **do**
    Sample 2 policies: $\pi_i, \pi_j \sim \Pi$;
    Sample K states: $s_1, s_2, \cdots, s_K \sim \mathcal{D}$;
    Take actions by both policies: $a_k^i \sim \pi_i(\cdot|s_k), a_j^n \sim \pi_j(\cdot|s_k)$;
    Construct a pairwise policy decision sequence:
    $\tau_{i,j} = <(s_1, a_1^i), (s_1, a_1^j), \cdots, (s_K, a_K^i), (s_K, a_K^j)>$;
    Encode state-action pairs into input tokens: $\mathbf{x}_k^i = f(s_k, a_k^i), \mathbf{x}_k^j = f(s_k, a_k^j)$;
    Add policy positional embeddings to tokens:
    $\mathbf{z}_0 = [\mathbf{e}_0, \mathbf{x}_1^i + \mathbf{e}_\alpha, \mathbf{x}_1^j + \mathbf{e}_\beta, \cdots \mathbf{x}_K^i + \mathbf{e}_\alpha, \mathbf{x}_K^j + \mathbf{e}_\beta]$
    Get the pairwise policy representation: $\chi_{i,j} = f(\mathbf{z}_0)$;
    Get the probability of whether $\pi_i$ performs better than $\pi_j$: $\hat{y}_{i,j} = o(\chi_{i,j})$;
    **for** *each OPE worker* $r \in \mathcal{W}$ **do**
        Generate the noisy labels: $y_{i,j}^r = \mathbb{1}_{\mathfrak{R}_i^r > \mathfrak{R}_j^r}$;
        Get the estimated noisy labels by crowd layer $\xi^r$: $\hat{y}_{i,j}^r = \xi^r(\hat{y}_{i,j})$;
        Compute the loss according to Equation (7);
        Backpropagate gradients to all parameters for minimizing the loss;
    **end**
    $N \leftarrow N - 1$;
**end**

---

# 5 EXPERIMENTS

In this section, we compare SOCCER with widely-used OPE methods on various tasks. We also present ablation studies with respect to PCT and CL, which are the main components of SOCCER.

## 5.1 EXPERIMENTAL SETTINGS

**Trajectory Set.** We evaluate SOCCER and all baseline OPE methods on D4RL dataset consisting of various trajectory sets (Fu et al., 2020). These sets of trajectory data are generated by different behavioral policies in different simulated environments. Overall, we adopt trajectory sets collected

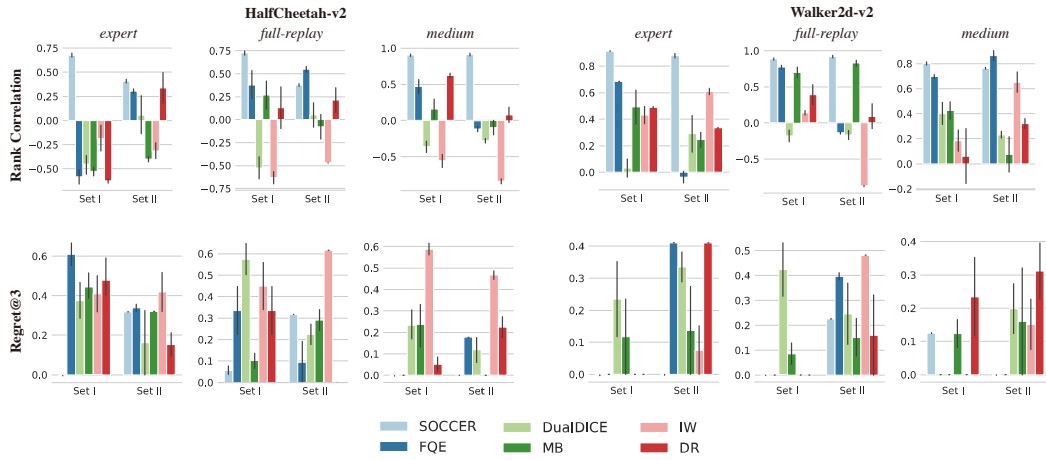

Figure 3: **Comparing SOCCER with other OPE baselines.**

from 2 environments of Mujoco games: `Hopper-v2` and `Walker2d-v2`. Besides, there are 3 different types of trajectory sets for each environment: `expert`, `full-replay` and `medium`. The difference between them is that the behavioral policies collecting these 3 types of trajectories show different performance in the simulated environment. And these behavioral policies are trained by the Soft Actor-Critic (SAC) algorithm online (Haarnoja et al., 2018).

**Policy Set.** To evaluate the abilities of all methods to correctly rank a set of policies in an offline way. We use the policy set released by Jin et al. (2022) as the candidate set of policies. For each trajectory set mentioned above, there are 2 types of policy sets ( referred as `Set I` and `Set II`) in which the expected return of policies are evenly spaced in the performance range of them. As mentioned in Jin et al. (2022), `Set I` and `Set II` aim to simulate two kinds of OPE cases. `Set I` consists of policies trained by offline RL algorithms (CQL (Kumar et al., 2020), BEAR (Kumar et al., 2019), CRR (Wang et al., 2020)). Since these algorithms have different network architectures and the generated policies are snapshots of models that stop training at different epochs, the policies contained in `Set I` show diverse behavioral performance. This is aligned with practical cases where the sources of policies are diverse and unknown. On the other hand, `set II` contains policies trained by SAC, which is the same as the algorithm of training the behavioral policies. Therefore, `Set II` corresponds to the practical OPE cases, such as production development and update. The updated policies share many common properties with the policies generating the trajectory data. In the experiments of ablations, we also compare some models that use extra training policy sets for providing supervised labels. These training sets are also released by (Jin et al., 2022), and policies in them are trained by SAC online.

**Baselines** [1]. We compare SOCCER with six state-of-the-art baselines. **i)** Fitted Q-Evaluation (`FQE` (Hoang et al., 2019)). It is a value-based OPE method, which learns a neural network to approximate the Q-function of the evaluated policy by temporal difference learning on the trajectory set. **ii)** Model-based estimation (`MB` (Paduraru, 2013)). It learns the dynamics model of environment, and estimates the expected return of evaluated policies by computing their average returns of Monte-Carlo rollouts in the model environment. **iii)** Weighted importance sampling (`IW` (Mahmood et al., 2014)). It leverages weighted importance sampling to correct the weight of the reward, regarding the collected trajectory data distribution to the data distribution of the evaluated policy. **iv)** `DualDICE` (Nachum et al., 2019). It also aims to achieve distribution correction yet without directly using importance sampling. It learns an estimation of the state-action stationary distribution for achieving distribution correction. **v)** Doubly Robust (`DR` (Jiang & Li, 2016)). It utilizes an unbiased estimator that leverages an estimated environment model to decrease the variance of the unbiased estimates

---

[1]We leverage a popular implementation of OPE algorithms: `https://github.com/google-research/google-research/tree/master/policy_eval`. It contains the first 5 baselines used in our paper

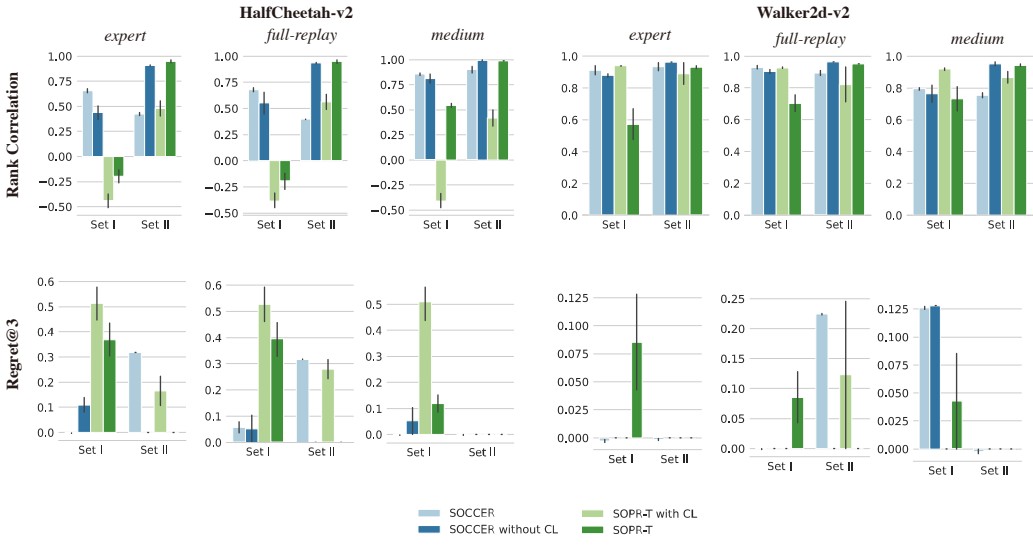

Figure 4: **Comparing SOCCER with other ablations.**

produced by importance sampling techniques. **vi**) SOPR-T [2] (Jin et al., 2022). This transformer-based model learns to achieve OPR tasks by learning a score function to score policies according to their generated state-action pairs. However, it relies on a strong assumption that there are extra deployed policies as well as these policies' true performance. Therefore, we only compare it when there are extra supervised information.

**Evaluation Metrics.** We evaluate all models according two widely-used metrics. **i**) Spearman's rank correlation. It is the Pearson correlation between the ground truth rank sequence and the evaluated rank sequence of the evaluated policies. **ii**) Normalized Regret@k. It is the normalized difference between the actual value of the best policy in the policy set, and the actual value of the best policy in the estimated top-k set. Mathematically, it can be computed by

$$regret@k = \frac{V_{max} - V_{topk}}{V_{max} - V_{min}},$$

where $V_{max}$ and $V_{min}$ is the expected return of the best and the worse policies, respectively, in the entire set, while $V_{topk}$ is the estimated top $k$ policies.

## 5.2 HYPERPARAMETERS.

In the implementation of our model, we control the scale of the learning objective function in our models by controlling the optimization procedure. It is conducted using Adam with a learning rate of $10^{-2}$, and with no momentum or weight decay. PCT is set as a 8-layer and 8-head transformer, and the dimensions of representations and all tokens are 128. The aggregation token $\mathbf{e}_0$ and the policy positional embeddings $\mathbf{e}_\alpha$ and $\mathbf{e}_\beta$ are initialized as random vectors. The token encoder $f$ is a 3-layer MLP followed by a ReLU() activation function, and its feedforward dimension is 128. The linear projection $\phi$ is a single-layer MLP. During training, we set the number of training epochs as 100, and use the model snapshots at the final epoch to achieve OPR tasks in experiments. For each epoch, we sample 1280 pairwise policy decision sequences for training, and each sequence contains 256 sampled states. Besides, in each epoch, we sample 5 models from the baseline models as the OPE worker to generate noisy labels. In the inference stage, we sample 32 pairwise sequences for each policy pairs to compute the average score of each policy. All experiments in this paper are carried out with 3 different seeds, and they are $1, 2, 3$, respectively.

---

[2] We use the official implementation of SOPR-T: https://github.com/SOPR-T/SOPR-T

### 5.3 Experimental Results

**Comparison with Other OPE Baselines.** We first compare SOCCER against 5 baselines across 2 environments. For each environment, there are 3 different trajectory sets in which each set contains 2 distinct policy sets. So there are 12 policy sets required to be correctly ranked by all models. We show the rank correlation and regret@k values of the estimated rank sequences generated by each model in Figure 3.

Overall, we can find SOCCER shows superior performance compared with other baselines. Furthermore, among all policy sets, SOCCER shows the highest rank correlations in 10 sets and the lowest regret@k in 8 sets, indicating that it can achieve robust and effective performance in diverse sets of policies. By contrast, the results of other baselines shows high variance across different policy sets. On the other hand, SOCCER shows great performance on learning noisy labels generated by other OPE workers. Note that in these experiments, all OPE workers which generate noisy labels for SOC-CER are directly sampled from the trained models of baselines. We can find that SOCCER could still perform well despite of the low quality of OPE workers. For example, looking at the results in `Set I` of the expert trajectory set in HalfCheetah-v2, all OPE baselines show poor performance (the average rank correlation of them is about $-0.27$), indicating that labels generated by them are very noisy. However, SOCCER shows a high performance (about $0.62$) while its supervised information is from such low-quality labels. Intuitively, we believe that there are two main reasons causing this phenomenon. First, the effective representation capability of our proposed policy comparison transformer could help to reduce the biases of noisy annotators. Second, the crowd layer could automatically distinguishes the good from the unreliable OPE annotators and captures their individual biases. In conclusion, SOCCER shows highly effective and robust OPR results across diverse kinds of policies. Besides, it can largely reduce the biases induced by OPE annotators and thus gets better performance than these OPE workers.

**Ablations.** To figure out the importance of each component in our framework. We perform several ablations on the same policy sets as the experiments mentioned above. All results are illustrated in Figure 4. Specifically, we compare SOCCER with 3 different models. The first one referred to `SOCCER without CL` is the model using our PCT architecture but discarding the crowd layer. Since it has no module to learn from crowds, so we make train it by providing extra sets of deployed policies released by (Jin et al., 2022). The second one is `SOPR-T` (Jin et al., 2022). It is also a transformer-based model to learn policy representations. The difference between it and SOCCER is that it learns an individual policy representation, while SOCCER learns pairwise policy representations which aim to capture subtle differences between any two policies. Furthermore, `SOPR-T` also cannot learn from crowds, so we train it in the same supervised way. As shown in Figure 4, `SOCCER without CL` shows better results than `SOPR-T` on 10 policy sets, and `SOCCER` performs better than `SOPR-T` on 8 policy sets. This indicates that in the OPR tasks, our proposed pairwise policy representation which aims to capture relations between intra-policy decisions has stronger representation ability than the pointwise policy representation, which regards each policy as an individual point, without considering the subtle decision differences between policies. On the other hand, `SOCCER` and `SOPR-T with CL` both show comparable performance with `SOCCER without CL` and `SOPR-T`, respectively, while `SOCCER` and `SOPR-T with CL` cannot get any supervised information from extra deployed policies. This indicates that regarding results estimated by other OPE methods as noisy labels and using a crowd layer to learn from such labels is an effective way when there is no sufficient extra policies providing supervised labels.

## 6 Conclusions

This paper proposes SOCCER, a novel self-supervised learning framework for addressing OPR problems. SOCCER is the first framework that combines crowdsourcing with OPR problems. In order to capture the relative discrepancies between policies, SOCCER employs a policy comparison transformer with a novel positional encoding to learn effective pairwise policy representations. Experimental results show that SOCCER not only outperforms baselines on various tasks but also generalizes well in diverse policy sets.

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
