# OpenReview forum: "Self-Supervised Off-Policy Ranking via Crowd Layer"
_ICLR.cc/2023/Conference — Submitted to ICLR 2023_

### Official Review · Reviewer_5DpY · 2022-10-22

**Confidence:** 4
**Correctness:** 3
**Technical Novelty And Significance:** 3
**Empirical Novelty And Significance:** 3
**Recommendation:** 6

**Clarity, Quality, Novelty And Reproducibility:**

- The paper is very well-written, and it's very easy to understand.
- The method proposed is novel and very practical, especially in a setting where access to deployed logging policies is limited, like clinical trials.


**Strength And Weaknesses:**

Strengths of the paper:

- The problem of OPE via policy ranking is very practical and bypasses the requirement for estimating the absolute policy value (expected reward), which is known to be a difficult problem.
- In practical settings, like recommender systems, ultimately we need a ranked list of policies, according to their estimated 'online' performance.
- The method can be used in a setting where access to multiple deployed policies with corresponding rewards is not feasible, for ex: in a clinical trial.
- Existing work from crowd-sourcing is used to learn from existing OPE methods while assuming they generate noisy labels, which is a fair assumption.

Some questions to the authors:
- How is a transformer-part important to the off-policy ranking pipeline? For ranking policies, have you compared with existing Learning-to-rank methods like LambdaMART on top of features extracted from trajectories of the policies, two-tower network (like Neural Matrix Factorization) with say RNN based feature extraction for both heads? For a given trajectory, a listwise method like lamdaMART can rank all the policies at a time, instead of a pairwise comparison.
- A clarification question (sorry if I missed something obvious): What is the dimensionality of $x_k$s (Eq. 2)? Is it a scaler, or a vector? Since you are adding a 2-d $e_\alpha$ vector to it, I am assuming it's a 2-D vector?
- In a setting like a recommender system/search, where you have access to a large log data with multiple policies and their rewards, how does this method compare with the baseline "SOPR-T"?

**Summary Of The Paper:**

The authors present an Off-Policy Evaluation (OPE) method geared toward ranking policies w.r.t their estimated performance, as compared to the standard task of predicting the estimated performance absolutely. It is more aligned with the OPE end goal, i.e. ranking policies among a set of candidates.

Previous work on off-policy ranking SOPR-T (Jin et al., 2022) used supervised learning for ranking policies, but it assumes access to multiple deployed policies with reward signals available. It is argued by the authors that this assumption is impractical, as it assumes multiple deployed policies along with their rewards in the log data.

Authors instead propose a self-supervised learning method, based on the 'learning from crowd' paradigm, where they assume different OPE methods as workers in a crowd-sourcing setting, with each worker generating noisy labels. A previous work, 'Crowd Layer (CL)' from learning via crowdsourcing literature is used to learn with the noisy label setting.

**Summary Of The Review:**

- Authors propose an off-policy ranking method, which directly predicts the ranking between two policies, using a ranking method.
- Assuming a setting where access to multiple logging policies is restricted, the proposed method can leverage existing OPE methods in a pseudo-crowd-sourcing setup, with each OPE method as a worker in the crowd, and learning from the workers in a noisy label setting.

---

> ### Author Response · Authors · 2022-11-19
> **Responses to Reviewer 5DpY**
>
> We thank the reivewer for your valuable comments and questions.
>
> $\\textbf{Q1}$: How is a transformer-part important to the off-policy ranking pipeline?
>
> $\\textbf{A1}$: Transformers are powerful due to its self-attention mechanism which can capture the correlations between input tokens. Moreover, the design of positional encoding makes Transformers work well on sequential data. In our Policy Comparision Transformer (PCT), the self-attention machanism successfully captures the correlation between state-action pairs, which is key to learn effective and generalizable policy representations. Since we want to learn pairwise policy representations, we design a novel positional encoding which reflects the order of two policies, so that the learned pairwise policy representations can be directly used to predict the order of them. To show the advantage of Transformers, we did additional ablation experiments by replacing the PCT with an MLP which has the same number of parameters. We present the average rank correlation of PCT and MLP of the 6-policy set in HalfCheetah-v2 environment as follows. We can see that PCT indeed performs much better MLP.
> $$\\begin{array}{|c|c|}
> \\hline\text{ } &\text{Rank Correlation}\\\\
> \\hline \text{PCT} &0.65\\\\
> \\hline \text{MLP} &0.32\\\\
> \\hline
> \\end{array}$$
>
> $\\textbf{Q2}$: For ranking policies, have you compared with existing Learning-to-rank methods?
>
> $\\textbf{A2}$: Actually, SOPR-T [1] can be regarded as a learning-to-rank method, which extracts features of two policies seperately and uses a loss function similar to RankNet to train the model.  SOPR-T is also the main baseline to our work (see Figure 4 in the paper for detailed comparisons).
>
> We agree with the reviewer that learning-to-rank is another way to approach the problem of off-policy ranking. To our knowledge, most learning-to-rank methods focus on designing appropriate loss function to learn good ranking results. However, in the problem of off-policy ranking, the policy representation plays an important role. Learning policy representations separately (as is done in SOPR-T and other learning-to-rank methods) does not capture the core discrepancy between two policies. Therefore, in this work we focus on learning pairwise policy representations to improve the policy ranking. There might be some oportunities for future work to combine pairwise representations with learning-to-rank methods.
>
>
> $\\textbf{Q3}$: A clarification question (sorry if I missed something obvious): What is the dimensionality of $x_k$ (Eq. 2)?
>
> $\\textbf{A3}$: Thanks for your question. Input token $x_k$ is a d-dimensional vector, where d is a hyper-parameter. One-hot vectors $e_{\\alpha}$ and $e_{\\beta}$ would also be mapped to d-dimensional vectors (positional encoding), so that they can be added to $x_k$. We will state more clearly in the paper.
>
> $\\textbf{Q4}$: Do you have access to a large log data with multiple policies and their rewards?
>
> $\\textbf{A4}$: Thanks for your advice. We actually have considered to test our method in recommender systems. Unfortunately, there are no public datasets that contains enough recommendation policies and their online performance. We note that SOPR-T does not include experiments on recommender systems either. We think that the access to this kind of data is the main obstacle. This is also the main motivation of our work: to develop solutions to off-policy ranking problems without ground-truth labels.
>
>
> [1] Supervised off-policy ranking. In ICML, 2022.

---

> > ### Comment · Reviewer_5DpY · 2022-11-20
> > **Response to authors comments**
> >
> > Thank you authors for the clarifications.
> >
> > A1: Thanks for the additional experimental result with an MLP. There seems to be a significant drop in performance. This does show the significance of using a transformer, although it would be more interesting to compare with a simpler sequence model like CNN/RNN, and compare the performance/run-time trade-off. Transforms can be costly in terms of inference time, a simple CNN/RNN might be faster. A suggestion for a future paper maybe :)
> >
> > A2: Thanks for your comments. I agree that the baseline SOPR-T can be considered an LTR method.
> >
> > A4: Thanks for the clarification.

---

> > > ### Author Response · Authors · 2022-11-21
> > > **Response to Reviewer 5DpY**
> > >
> > > Thank you for your kind suggestion!
> > >
> > > We will definitely consider to develop more efficient architectures for learning policy representations.

---

### Official Review · Reviewer_RMFP · 2022-10-24

**Confidence:** 4
**Correctness:** 3
**Technical Novelty And Significance:** 3
**Empirical Novelty And Significance:** 3
**Recommendation:** 3

**Clarity, Quality, Novelty And Reproducibility:**

The paper is generally well written and easy to follow, especially its core idea of converting off-policy evaluation into a binary classification problem. Every step in the proposed solution is standard, e.g., the crowd-layer for integrating different off-policy evaluators, and transformer to encode a sequence of actions. Hence, it should not be difficult to reproduce the algorithm pipeline. The authors also provided details about most of the hyper-parameter settings in the paper, which should help ensure the reproducibility of the reported results.

**Strength And Weaknesses:**

Strength:
+ The idea of off-policy rank evaluation is reasonable and valid, which to my best knowledge is first time discussed in literature. Although only high-level intuitive arguments are provided regarding the advantage of this new objective, it is still nice to observe its effectiveness in the reported empirical studies.
+ The reported empirical studies provide a comprehensive picture about the comparison between the proposed solution and a rich set of baseline methods.

Weakness:
- The proposed method lacks necessary theoretical justification, with unknown properties about the provided estimations, e.g., there is no way to quantify its bias and variance.
- The solution boils down to a binary classification problem, under which the crowd-sourcing idea is a natural extension when only noisy labels exist. Following this logic, methods for learning from noisy labels can also be leveraged to address the problem.
- The empirical nature of the proposed solution creates a large set of hyper-parameters, which make the tuning and comparison hard to exhaust. For example, presumably the number of states where we execute the policies to obtain their actions, the number of pairwise policy comparisons, and the number of OPE workers are important for the performance of the proposed solution. But there is no experiment evaluating the impact from such hyper-parameters.

Questions:
- In Section 5.2, it is mentioned that in each epoch, 5 OPE workers are randomly selected from the baseline models. I am not sure why we should sample different OPE workers every epoch. Shouldn’t they be the same throughout the training and testing stages for the crowd layer to learn the (equivalent) confusion matrix?
- What’s the principle to determine the number of states where we execute the policies to obtain their actions, the number of pairwise policy comparisons, and the number of OPE workers in practice? Are they the more the better?
- Why not have a dedicate policy encoder to represent each policy and then compare their embeddings for binary classification? Or we can simply follow the way we use transformer to encode two sentences into one embedding to embed the two policies. The conventional position embedding in transformer can help us realize the corresponding positions are actions from two policies but under the same states.
- My understanding about $e_\alpha$ and $e_\beta$ is that they are fixed one-hot vectors. But Section 5.2 described them as random vectors at initialization. Clarification is necessary here.


**Summary Of The Paper:**

This paper studies the problem of offline policy evaluation. The key idea is that instead of predicting the value of the target policy, the authors proposed to evaluate the relative rank of the policy, among a set of candidate policies. When there is no sufficient historical observations of different behavior policies, the authors proposed to leverage the idea of crowd-sourcing to integrate prediction from multiple offline policy evaluators. The final ranking/scoring of each target policy is obtained by aggregating the pairwise comparisons among all candidate policies. Experiments on a set of simulated offline trajectory data were performed against a rich collection of offline policy evaluation methods.

**Summary Of The Review:**

The idea of studying offline policy ranking, instead of evaluation, is an interesting and practical idea. But the proposed solution is overly simplified and lacks necessary theoretical justification or analysis. It is hard to know when the algorithm would work better than standard off-policy evaluation methods.

---

> ### Author Response · Authors · 2022-11-19
> **Responses to Reviewer RMFP (Part 2)**
>
> $\\textbf{Q5}$: Why we should sample different OPE workers every epoch. Shouldn’t they be the same throughout the training and testing stages for the crowd layer to learn the (equivalent) confusion matrix?
>
> $\\textbf{A5}$: In implementation, we would equip each worker with an independent crowd layer (2 parameters, the scale factor and the bias). In each epoch, only the crowd layers corresponding to the sampled workers would be trained. In the testing stage, all crowd layers would be removed, and the policy comparison can be directly predicted by the rest architecture of our model.
>
> $\\textbf{Q6}$: The empirical nature of the proposed solution creates a large set of hyper-parameters, which make the tuning and comparison hard to exhaust.
>
> $\\textbf{A6}$: In our framework, most hyper-parameters are brought by the existing OPE methods. However, many existing OPE methods are open-sourced and well-tuned. Therefore, it is convenient to reproduce them in practice.
>
> In addition, our framework includes two types of hyper-parameters, we will show how to select these hyper-parameters as follows.
>
> (1) Hyper-parameters of the model architecture (such as the depth of transformer and the learning rate). For these hyper-parameters, we actually use grid-search to determine the best choices of them in our experiments. We will state more clearly in the paper.
>
> (2) The batch size of state-action pairs. During training, we feed a batch of state-action pairs to the Transformer in order to compute the approximated policy representations. To balance the computational cost and the performance, we chose the number 256 as the batch size. This choice is supported by the experimental results bellow, which show the averaged rank correlations of our model with the batch size growing. We can find that when the batch size is larger than 256, the performance of our model tends to be stable.
>
> $$\\begin{array}{|c|c|c|c|c|c|c|c|c|c|}
> \\hline\text{Batch Size} &8	&16	&32	&64&	128	&256&	512&	1024&	2048\\\\
> \\hline \text{Avg. Rank Correlation} &0.21&	0.27&	0.37&	0.39&	0.56&	0.65	&0.64&	0.66	&0.65\\\\
> \\hline
> \\end{array}$$

---

> ### Author Response · Authors · 2022-11-19
> **Responses to Reviewer RMFP (Part 1)**
>
> We thank the reviewer for your constructive comments.
>
> $\\textbf{Q1}$: The proposed method lacks necessary theoretical justification, with unknown properties about the provided estimations, e.g., there is no way to quantify its bias and variance.
>
> $\\textbf{A1}$: Since our framework combines the existing OPE methods and crowdsourcing techniques, we also enjoy the theoretical gurantees of these two fields of research. Intuitively, we can choose high-quality OPE workers whose performance have been demonstrated emperically or theoretically.
>
> For example, DualDICE [1] and FQE [2] present that the error associated with their estimated policy performance can be theoretically bounded under some mild assumptions. And doubly-robust method [3] proves that its estimated results enjoy reduced variance when the value estimator is accurate. These theoretical analysis provide guidlines on selecting high-quality OPE workers. On the other hand, some works from the crowdsourcing field provide theoretical guarantees on the learning performance. For example, [4] proves that as long as the worker qualities exceed a threshold, the generalization error of the trained model is bounded. This result builds the connection between worker quality and the final learning performance. We agree with the reviewer that a unified theoretical quarantee would further improve the reliability of our framework. However, we want to emphasize that our work provides the first practical crowdsourcing based framework to address the off-policy ranking problem.
>
> $\\textbf{Q2}$: The solution boils down to a binary classification problem, under which the crowd-sourcing idea is a natural extension when only noisy labels exist. Following this logic, methods for learning from noisy labels can also be leveraged to address the problem.
>
> $\\textbf{A2}$: We agree with the reviewer that other learning from crowd (noisy labels) methods can also be leveraged to address the problem. We regard it as a merit because any future advances in the field of crowdsourcing could benefit the problem of off-policy ranking. In this work, we choose to use the Crowd Layer because it is naturally compatible with deep learning approaches and can be trained in an end-to-end way. We also add some experiments to compare CL with another two label aggregation methods, namely "Average Score" and "Majority Voting". We report their performance on rank correlations in the following table. We can see from the following table that our method domindates these two baselines in all of the six environements.
>
> $$\\begin{array}{|c|c|c|c|}
> \\hline\text{Environment} &\text{Avg. Score}&\text{Major Voting}&\text{Ours}\\\\
> \\hline HalfCheetah-expert &-0.34	&-0.27	&\textbf{0.71}\\\\
> \\hline HalfCheetah-full-replay &0.24&	0.31&	\textbf{0.74}\\\\
> \\hline HalfCheetah-medium &0.32&	0.57&	\textbf{0.81}\\\\
> \\hline Walker2d-expert &0.53	&0.23&	\textbf{0.85}\\\\
> \\hline Walker2d-full-replay &0.41&	0.31&	\textbf{0.82}\\\\
> \\hline Walker2d-medium &0.21&	0.29&	\textbf{0.80}	\\\\
> \\hline
> \\end{array}$$
>
> $\\textbf{Q3}$: Why not have a dedicate policy encoder to represent each policy and then compare their embeddings for binary classification? Or we can simply follow the way we use transformer to encode two sentences into one embedding to embed the two policies. The conventional position embedding in transformer can help us realize the corresponding positions are actions from two policies but under the same states.
>
> $\\textbf{A3}$: The idea of using a dedicate policy encoder is used in SOPR-T [5], which serves as the main baseline to our method. Moreover, if we understand correctly, the method that the reviewer describes exactly matches our design of the Policy Comparison Transformer. In PCT, we utilize the self-attention mechanism of transformer to directly learn pairwise policy representations, which is proved to be more effective than SOPR-T. Detailed experimental results can be found in Figure 4 of our paper.
>
> $\\textbf{Q4}$: My understanding about $e_{\\alpha}$ and $e_{\\beta}$ is that they are fixed one-hot vectors. But Section 5.2 described them as random vectors at initialization. Clarification is necessary here.
>
> $\\textbf{A4}$: $e_{\\alpha}$ and $e_{\\beta}$ represent one-hot vectors that indicate the orders of two policies. In practice, $e_{\\alpha}$ and $e_{\\beta}$ will be further mapped to learnable embeddings whose dimensions match the input tokens (recall that positioned encodings are added to input tokens). We will state more clearly in the next version of our paper.
>
> [1] DualDICE: Behavior-agnostic estimation of discounted stationary distribution corrections. In NeurIPS, 2019.
>
> [2] Batch policy learning under constraints. In ICML, 2019.
>
> [3] Doubly robust off-policy value evaluation for reinforcement learning. In ICML, 2016.
>
> [4] Learning from noisy singly-labeled data. In ICLR, 2018.
>
> [5] Supervised off-policy ranking. In ICML, 2022.

---

> > ### Comment · Reviewer_RMFP · 2022-11-23
> > **Theoretical analysis**
> >
> > I am afraid I cannot be convinced that because both OPE and crowd sourcing have theoretical results, the proposed solution will have theoretical results. For example, the references on OPE mentioned in the paper and rebuttals are all about off-policy evaluation problem, i.e., evaluating the value of a target policy. Since this paper is about the rank of policies, I do not see an easy connection between those existing solutions to the proposed one in this paper. It would be helpful if the authors could shed light on it.

---

> > > ### Author Response · Authors · 2022-12-10
> > > **Theorem 1: OPE methods who have lower biases are high-quality workers for OPR**
> > >
> > > We are sorry for the late reply. We thank the reviewer for your useful suggestion. Following the reviewer's suggestion, we provide the following theoretical analysis to futher demonstrate the superiority of our method.
> > >
> > > Intuitively, if an OPE method could accurately predict two policies' online performances, it would be regarded as a high-quality worker in the crowdsourced OPR task. We formally define the worker quality as follows.
> > >
> > > **Definition 1** (OPE worker quality)
> > > Given two policies $\pi_i$ and $\pi_j$, we denote by $\delta_i$ and $\delta_j$ their real expected returns. For an OPE worker $r$, we denote by  $b^r_i$ and $b^r_j$ the prediction biases of $r$. The quality $\eta^r$ of the OPE worker $r$ is defined as the probability that $ y^r_{i,j}$ equals to $ y_{i,j}$:
> > >
> > > $$\qquad \qquad \qquad \eta^r =  P( y_{i,j} = y^r_{i,j})$$
> > >
> > > where $y_{i,j} = \mathbb{1_{\delta_i > \delta_j}}$ denotes the ground-truth label of whether $\pi_i$ performs better than $\pi_j$ and $ y^r_{i,j} = \mathbb{1_{\delta_i +b^r_i > \delta_j +b^r_j}}$ denotes the label predicted by worker $r$.
> > >
> > > From this definition we can see that the worker quality is actually determined by the biases. We derive the following theorem to reveal the connection between biases and worker quality.
> > >
> > > **Theorem 1** (Close-form representation of worker quality.) Assume that the bias of worker $r$ is bounded: $|b^r| \le B_r$, and $b^r$ subjects to a uniform distribution $U(-B_r, B_r)$. We denote by $\Delta = |\delta_i - \delta_j|$ the distance between the real expected returns of two policies. Then we have
> > > \begin{equation}
> > >    \qquad \qquad \qquad \eta^r = \frac{1}{2} +\frac{\Delta}{4B_r} \quad if \quad \Delta <2B_r \quad and \quad \eta^r =1 \quad otherwise.
> > > \end{equation}
> > >
> > > In fact, $\Delta$ is a fixed number if two policies are given. **Therefore, this theorem indicates that the smaller the bias bound $B_r$, the higher the worker quality $\eta^r$.**
> > >
> > >
> > > **Proof of Theorem 1.** Given $\pi_i$ and $\pi_j$, recall that $\eta^r = P( y_{i,j} = y^r_{i,j})$, we have:
> > >
> > > $$\qquad \qquad \qquad \eta^r =  P( y_{i,j} = y^r_{i,j}) = P(\mathbb{1_{\delta_i > \delta_j}} = \mathbb{1_{\delta_i +b^r_i > \delta_j +b^r_j}})$$
> > > $$\qquad \qquad \qquad \quad = P[(\delta_i - \delta_j)(\delta_i +b^r_i - \delta_j -b^r_j)>0]$$
> > > $$\qquad \qquad \qquad \quad= P[(\delta_i - \delta_j)>0] \cdot P[(b^r_i-b^r_j)>-(\delta_i-\delta_j)]$$
> > > $$\qquad \qquad \qquad \quad\quad + P[(\delta_i - \delta_j)<0] \cdot P[(b^r_i-b^r_j)<-(\delta_i-\delta_j)]$$
> > >
> > >
> > > Let $\zeta = b^r_i-b^r_j$, we have $\eta^r = P(\zeta>-\Delta)$ if $\delta_i - \delta_j >0$, and $\eta^r = P(\zeta<\Delta)$ if $\delta_i - \delta_j < 0$.
> > > Recall that $b^r \sim U(-B_r,B_r)$ and $b_i^r$, $b_j^r$ are independent from each other, then random variable $\zeta$ follows a triangular distribution whose probability density function can be written as:
> > >
> > > $$\qquad \qquad \qquad \quad f(\zeta) = \frac{1}{2B_r} + \frac{1}{4B^{2}_r}\cdot \zeta \qquad if  \quad -2B_r \le \zeta < 0$$
> > > $$\qquad \qquad \qquad \quad f(\zeta) = \frac{1}{2B_r} - \frac{1}{4B^{2}_r}\cdot \zeta \qquad if \quad 0 \le \zeta \le 2B_r$$
> > > $$\qquad \qquad \qquad \quad f(\zeta) = 0 \qquad otherwise $$
> > >
> > > With this close-formed density function,  the worker quality can be calculated as $\eta^r = \int_{-\Delta}^{+\infty}f(\zeta)d\zeta$ if $\delta_i - \delta_j >0$, and $\eta^r = \int_{-\infty}^{\Delta}f(\zeta)d\zeta$ if $\delta_i - \delta_j <0$. As it is easy to see that $\int_{-\Delta}^{+\infty}f(\zeta)d\zeta = \int_{-\infty}^{\Delta}f(\zeta)d\zeta$, we finally have
> > >
> > >    $$\qquad \qquad \qquad \quad  \eta^r = \frac{1}{2} +\frac{\Delta}{4B_r}  \qquad if \quad \Delta <2B_r$$
> > >    $$\qquad \qquad \qquad \quad  \eta^r = 1 \qquad \qquad \qquad if \quad \Delta \ge 2B_r$$

---

> > > > ### Author Response · Authors · 2022-12-10
> > > > **Theorem 2: Upper bound on the emperical risk**
> > > >
> > > > **Theorem 1** suggests that we can select high-quality OPE workers who have small biases. Now we show how the worker qualities influence the final performance of SOCCER.
> > > >
> > > > Recall that our input data can be represented by $\tau_{i,j}=<(s_1,a_1^i),(s_1,a_1^j),...,(s_K,a_K^i),(s_K,a_K^j)>$, which contains the different actions generate by $\pi_i$ and $\pi_j$ at the same set of states. We denote by $y_{i,j}$ the ground-truth label indicating whether $\pi_i$ performs better than $\pi_j$. We also denote by $y^1_{i,j},\cdots,y^M_{i,j}$ the labels provided by $M$ OPE workers. Then the ground-truth dataset (which we cannot access in practice) and the noised dataset can be represented as $\mathcal{D} =$ {$(\tau_{i,j}, y_{i,j})$} and $\mathcal{\widetilde D} =$ {$(\tau_{i,j}, y^1_{i,j},\cdots,y^M_{i,j})$}, respectively.
> > > >
> > > > Given an loss function $\ell$, we define the empirical risk for the given learner $\hat g \in \mathcal{G}$ learning with ground truth labels as $R_{\mathcal{D}}(\hat g) = \mathbb{E_{\mathcal{D}}} \Big[\ell(\hat g(\tau_{i,j}),y_{i,j})\Big]$, and the empirical risk learning with crowd layer as: $R_{\mathcal{\widetilde D}}(\hat g) = \frac{1}{M}  \sum_{r=1}^M \mathbb{E_{\mathcal{ D}^r}}\Big[\ell(\hat g(\tau_{i,j}), y^r_{i,j})\Big] $. The following theorem gives an upper bound on the difference between $R_{\mathcal{D}}(\hat g)$ and $R_{\mathcal{\widetilde D}}(\hat g)$.
> > > >
> > > > **Theorem 2** (Upper bound on emperical risk.)  Assume that for a given dataset $D$ and any $g \in \mathcal{G}$, $R_{\mathcal{D}}(g)$ is upper bounded by $\overline \ell$ and lower bounded by $\underline \ell$. Denote by $\widetilde \eta =\frac{1}{M}  \sum_{r=1}^M \eta^r$. Then for any given learner $\hat g$, we have
> > > >
> > > > $$\qquad \qquad \qquad R_{\mathcal{D}}(\hat g) - R_{\mathcal{\widetilde D}}(\hat g) \le (1-\widetilde \eta)\cdot (\overline \ell - \underline \ell).$$
> > > >
> > > > Moving $R_{\mathcal{\widetilde D}}(\hat g)$ to the right side we would have an upper bound on $R_{\mathcal{D}}(\hat g)$. **Although we cannot directly minimize  $R_{\mathcal{D}}(\hat g)$ due to the lack of ground-truth labels, this theorem suggests that we can minimize its upper bound instead. Moreover, choosing high-quality workers leads to a higher $\widetilde \eta$ and a tighter upper bound of $R_{\mathcal{D}}(\hat g)$.**
> > > >
> > > > **Proof of Theorem 2.** We start by expanding $R_{\mathcal{D}}(\hat g) - R_{\mathcal{\widetilde D}}(\hat g)$:
> > > >
> > > > \begin{equation}
> > > >   \qquad \quad \quad R_{\mathcal{D}}(\hat g) - R_{\mathcal{\widetilde D}}( \hat g) = \mathbb{E_{\mathcal{D}} }\Big[\ell(\hat g(\tau_{i,j}),y_{i,j})  \Big] - \frac{1}{M}  \sum_{r=1}^M\mathbb{E_{\mathcal{D^r}}} \Big[\ell (\hat g(\tau_{i,j}),y^r_{i,j})  \Big]
> > > > \end{equation}
> > > > $$\qquad \qquad \qquad \quad = \mathbb{E_{\mathcal{D}}} \Big[\ell(\hat g(\tau_{i,j}),y_{i,j})  \Big]- \frac{1}{M}  \sum_{r=1}^M \mathbb{E_{\mathcal{D^r}}} \Big[\ell(\hat g(\tau_{i,j}), y^r_{i,j})\Big]$$
> > > > $$\qquad \qquad \qquad \quad = \frac{1}{M} \sum_{r=1}^M \Bigg[\mathbb{E_{\mathcal{D}}}\Big[ \ell(\hat g(\tau_{i,j}),y_{i,j}) \Big] -  \mathbb{E_{\mathcal{D^r}}}\Big[\ell(\hat g(\tau_{i,j}), y^r_{i,j})\Big]\Bigg]$$
> > > > $$\qquad \qquad \qquad \quad \le \max_{g \in \mathcal{G}}\frac{1}{M} \sum_{r=1}^M \Bigg|\mathbb{E_{\mathcal{D}}}\Big[ \ell( g(\tau_{i,j}),y_{i,j}) \Big] -  \mathbb{E_{\mathcal{D^r}}} \Big[\ell( g(\tau_{i,j}), y^r_{i,j})\Big]\Bigg|$$
> > > > $$\qquad \qquad \qquad \quad =  \max_{g \in \mathcal{G}}\frac{1}{M} \sum_{r=1}^M \Bigg|\mathbb{E_{\mathcal{D}}}\Big[ \ell( g(\tau_{i,j}),y_{i,j}) \Big] -  \mathbb{E_{(\pi_i,\pi_j)\sim \mathcal{ D^r},y^r_{i,j}=y_{i,j}}} \Big[\ell( g(\tau_{i,j}), y_{i,j})\Big]$$
> > > > $$\quad \qquad \qquad \qquad \quad-\mathbb{E_{(\pi_i,\pi_j)\sim \mathcal{ D^r},y^r_{i,j}=1-y_{i,j}}} \Big[\ell( g(\tau_{i,j}),1-y_{i,j})\Big] \Bigg|$$
> > > > $$\qquad \qquad \qquad \quad =  \max_{g \in \mathcal{G}}\frac{1}{M} \sum_{r=1}^M \Bigg|\mathbb{E_{\mathcal{D}}}\Big[ \ell( g(\tau_{i,j}),y_{i,j}) \Big] -  \mathbb{E_{\mathcal{D}}} \Big[\eta^r\cdot\ell( g(\tau_{i,j}), y_{i,j})\Big]$$
> > > > $$\qquad \qquad \qquad \quad \quad-  \mathbb{E_{\mathcal{D}}} \Big[(1-\eta^r)\cdot\ell( g(\tau_{i,j}),1-y_{i,j})\Big] \Bigg|$$
> > > > $$\qquad \qquad \qquad \quad =  \max_{g \in \mathcal{G}}\frac{1}{M} \sum_{r=1}^M \Bigg|\mathbb{E_{\mathcal{D}}}\Big[(1-\eta^r) (\ell( g(\tau_{i,j}),y_{i,j}) - \ell( g(\tau_{i,j}),1-y_{i,j}))\Big]  \Bigg|$$
> > > > $$\qquad \qquad \qquad \quad \le (1-\widetilde \eta)\cdot( \mathcal{\overline \ell} -  \underline \ell)$$

---

> > > > > ### Comment · Reviewer_RMFP · 2022-12-12
> > > > > **Will definitely carefully read the new analysis**
> > > > >
> > > > > I really appreciate the authors' effort in providing the new theoretical analysis about the proposed algorithm.
> > > > >
> > > > > As I was out of town during the weekend, I need sometime to carefully digest the new results. In the meanwhile, I understand the discussion phase is coming to an end today, and thus I will try my best to follow up as soon as possible (within 24 hours).
> > > > >
> > > > > Just a few quick clarification questions that help me better understand the results:
> > > > > 1. what is a leaner $g\in\mathcal{G}$ in Theorem 2? $g$ in the paper is defined as a mapping from logged trajectory to $n$-dimensional representation.
> > > > > 2. how do we know the bias of each OPE method?
> > > > > 3. Theorem 1 is about each OPE's bias and Theorem 2 is about the quality of $g$ learning. How are these two related?
> > > > >
> > > > > Thanks!

---

> > > > > > ### Author Response · Authors · 2022-12-12
> > > > > > **Thank you for your precious time**
> > > > > >
> > > > > > We really appreciate that you will spend your precious time in checking our results. Please find the clarifications bellow.
> > > > > >
> > > > > > 1. The learner $g$ in the proof actually represents the model (including the Crowd Layer) that maps trajectories $\tau_{i,j}$ to labels $y_{i,j}$. The notation is slightly abused. We will explain more clear in the final version of the paper.
> > > > > >
> > > > > > 2. It is hard to accurately quantify the biases of the OPE methods. However, existing works have provided upper bounds on the biases [1,2], which we denote as $B_r$ in Theorem 1.
> > > > > >
> > > > > > 3. Theorem 2 provides an upper bound on the emperical risk of learner $g$. We can see from the results that this bound depends on the averaged worker quality $\widetilde \eta$. Recall that Theorem 1 suggests that we can select workers with small biases to improve the worker qualities. Combine them together we can conclude that OPE workers with small biases lead to a tighter upper bound on the emperical risk, thus a better learning performance in OPR task.
> > > > > >
> > > > > > We will be ready to address your further concerns. Please feel free to ask us if clarifications are needed. Thanks again!
> > > > > >
> > > > > > [1] Batch policy learning under constraints. In ICML, 2019.
> > > > > >
> > > > > > [2] DualDICE: behavior-agnostic estimation of discounted stationary distribution corrections. In NeurIPS, 2019.

---

> > > > > > > ### Comment · Reviewer_RMFP · 2022-12-13
> > > > > > > **The proofs do not really address the concern**
> > > > > > >
> > > > > > > Finally I got some time to digest the newly added proofs and here are my understandings.
> > > > > > >
> > > > > > > The proofs are basically applying what's known for analyzing crowdsourcing onto the proposed crowdsourcing-based off-policy ranking problem. As a result, the Theorem 2 shows we can optimize the upper bound of the empirical risk of a learner over a set of OPEs. Such a result is fine for learning from crowdsourced labels, where we care about the quality of the learnt classifier. But in OPE, we care about policy evaluation, or more precisely policy ranking in this paper. I do not think Theorem 2 answers this question. For example, how does Theorem 2 suggest the probability that the proposed method finds the truly best policy?
> > > > > > >
> > > > > > > Theorem 1 is also problematic: it is also borrowed from crowdsourcing field, where people often assume independent annotators and each annotator makes independent mistakes. But as annotators become specific OPE methods in this work, they are not necessarily independent from each other, e.g., they might be different types of model-based or IPS estimators. And the assumption that within a particular OPE method biases towards different policies are uniformly random is also very strong.
> > > > > > >
> > > > > > > As a result, I do not think the provided proofs address the real need of theoretical analysis in this paper.

---

> > > > > > > > ### Author Response · Authors · 2022-12-13
> > > > > > > > **There are some misunderstandings**
> > > > > > > >
> > > > > > > > We thank the reviewer for your time. However, we feel that you might have some misunderstandings on our theorems.
> > > > > > > >
> > > > > > > > 1. **Relations between emperical risk and policy ranking.** The logic is: the lower the emperical risk, the more accurate the classifier (which predicts whether a policy is better than the other). If the classifier is accurate, we can easily select the best policy out of a set of policies. In other words, the OPR task is reduced to learning an accurate classifier.
> > > > > > > >
> > > > > > > > 2. **The workers are independent from each other.** When we judge whether two OPE methods are independent from each other, we only need to judge whether the learning process of one worker depends on the other. Actually, although they share the same set of training data, the learning process of the workders are indeed independent, no matther whether the basic learning methods are similar.
> > > > > > > >
> > > > > > > > 3. **The distribution of biases.** There are many ways to describe the distribution of biases, among which the uniform distribution is the simplest version. If we assume other kinds of distributions of biases, we can still get close-form representations or upper bounds on the biases. We will leave it for future work.
> > > > > > > >
> > > > > > > > We hope that the above replies address the reviewer's concerns. Thank you!

---

> > > > > > > > > ### Comment · Reviewer_RMFP · 2022-12-13
> > > > > > > > > **Not my misunderstandings but mistakes in the follow-up explanations**
> > > > > > > > >
> > > > > > > > > I am afraid this time I cannot agree with most of the follow-up explanations and believe they are incorrect.
> > > > > > > > >
> > > > > > > > > 1. The intuition is correct, i.e., the lower the empirical risk of learning from crowdsourced labels is, the more accurate OPR is. But there is no direct translation from empirical risk to OPR in Theorem 2. For example, how do we relate the regret of OPR selection, i.e., whether the best policy is identified, with the empirical risk or Theorem 2? I believe in both OPR and OPE, we care whether we are choosing the best policy.
> > > > > > > > > 2. Independent OPE training does not lead to independent OPE results across different estimators, which is the key in Theorem 2. Let me be more specific. First, to give you a naive example: for a given training set for a classification problem, we can independently train a SVM classifier and a logistic regression classifier; but their classification results are not necessarily independent from each other, as they are trained on the same set of instances. It is the same issue here for a set of OPEs. Second, in Theorem 2, the summation over $M$ OPEs already assumes they independent. This is the common assumption in crowdsourcing, on which your proof is based.
> > > > > > > > > 3. The distribution of biases of an OPE clearly depends on specific OPE methods. For example, IPS is known to be unbiased if ground-truth propensity score is known; otherwise, its bias totally depends on the estimated propensity. I cannot understand why/how we can assume how the bias is distributed.

---

> > > > > > > > > > ### Author Response · Authors · 2022-12-13
> > > > > > > > > > **We believe our theorems and proofs are correct. Misunderstandings lie in what you expect from the theorems.**
> > > > > > > > > >
> > > > > > > > > > 1. Theorem 2 only gives an formal upper bound on the empirical risk. Given the intuition that the lower the empirical risk, the more accurate the classifier, we are expected to find the best policy with a high probability.  The reviewer seems to ask us to reveal the connection between the risk and the accuracy, which we think is unnecessary. Taking the simple SVM as example, we do not care the exact relations between risk the accuracy, instead, we just minimize the risk to improve the accuracy.
> > > > > > > > > >
> > > > > > > > > > 2. We define the independence of two workers as the independence of their prediction results (as well as the incured loss). In fact, one worker's predictions depend only on the inputs and its own loss function, which are irrelevant of other workers' prediction results. In other words, changing other workers' predictions (or hyper-parameters, loss functions) does not influence the target worker's predictions and losses, no matter whether they are using the same set of training data.
> > > > > > > > > >
> > > > > > > > > > 3. We agree that different OPE methods have different biases. Since there might be many different OPE methods, we cannot use one theorem to cover them all. As we have explaned, we only provided a special case where the biases are uniformly distributed. Theorem 1 might be in different forms under different assumptions on the distribution of the biases.
> > > > > > > > > >
> > > > > > > > > > We do appreciate the reviewer's efforts in checking our results. We sincerely hope that our responses help to solve the issues.

---

> > > > > > > > > > > ### Comment · Reviewer_RMFP · 2022-12-14
> > > > > > > > > > > **Maybe these references would be helpful**
> > > > > > > > > > >
> > > > > > > > > > > Let me first share some references that should help the authors understand what we care about in OPE's theoretical analysis:
> > > > > > > > > > >
> > > > > > > > > > > 1. Doubly Robust Off-policy Value Evaluation for Reinforcement Learning: http://proceedings.mlr.press/v48/jiang16.pdf
> > > > > > > > > > > 2. Doubly Robust Policy Evaluation and Learning: https://arxiv.org/pdf/1103.4601.pdf
> > > > > > > > > > > 3. DualDICE: Behavior-Agnostic Estimation of Discounted Stationary Distribution Corrections: https://proceedings.neurips.cc/paper/2019/file/cf9a242b70f45317ffd281241fa66502-Paper.pdf
> > > > > > > > > > >
> > > > > > > > > > > Basically, OPE aims to evaluate the return/value of a policy, and thus we should analyze the bias and variance/MSE of the evaluation of an OPE estimator. Then, in this work, since a new principle named OPR is proposed, what should be analyzed is the quality of policy ranking. Using your own experiments as example/evidence, rank correlation and regret were treated as the main metric, because the proposed method is designed to rank/select the best policy.
> > > > > > > > > > >
> > > > > > > > > > > Again, directly borrowing proof from learning from crowdsourced labels does not work, as these two problems care distinct criteria.
> > > > > > > > > > >
> > > > > > > > > > > Regarding point 2, I encourage the authors to consider the setting in my earlier post: hint - consider whether the two classifiers can still make independent mistakes.
> > > > > > > > > > >
> > > > > > > > > > > Regarding point 3, I am afraid we just cannot assume how an OPE estimator's bias distributes, otherwise all the references I provided would be meaningless.

---

### Official Review · Reviewer_KHEq · 2022-10-24

**Confidence:** 4
**Correctness:** 3
**Technical Novelty And Significance:** 2
**Empirical Novelty And Significance:** 1
**Recommendation:** 6

**Clarity, Quality, Novelty And Reproducibility:**

The overall quality of this paper as well as the reproducibility of the proposed method is fair. The proposed method is technically sound. However, the novelty is relatively limited. From my point of view, the proposed method is like a combination of several existing methods in a simplified manner.

**Strength And Weaknesses:**

Strength:
1. This paper simplifies the OPE into OPR tasks.
2. This paper proposes an end-to-end method to solve the OPR.
3. The proposed method is relatively technical sound.

Weaknesses:
1. The experiment settings lack some explanations, e.g. the reason for the specific value of hyper-parameters.
2. The figures need to be improved, e.g. sub-figures in Fig.4 are not aligned.
3. Ablation experiments as well as baselines need to be considered more carefully. From my point of view, the essence of the proposed method is similar to a label aggregation method with deep learning. Thus, some other structure-like method should also be added, e.g. SpeeLFC. The ablation experiment introduces an extra strategy to obtain ‘truth’ to train the method. This setting lowers the convincing performance of the proposed method.


**Summary Of The Paper:**

To fulfill OPE task, this paper simplifies OPE into OPE and proposes an end-to-end method, namely SOCCER. SOCCER is compared with several baselines in two environments. Experimental results show that the proposed method achieves high accuracy with certain hyper-parameters. An ablation experiment is also performed.

**Summary Of The Review:**

This paper simplifies the OPE task into an OPR task and proposes an end-to-end method, namely SOCCER, to tackle the problem. SOCCER contains policy representation (pairwise), a feature extractor (MLP and MSA), and a training strategy (Crowd Layer). SOCCER is compared with several baselines in two environments with three modes. The process of the experiment is relatively complete. An ablation experiment is also performed, while some settings make the results not very convincing. Experimental results show that the proposed method can achieve good performance in some conditions. Although this work lacks theoretical innovation and the representation of this paper has some flaws, the entire work is relatively complete and the OPE problem is relatively solved.

---

> ### Author Response · Authors · 2022-11-19
> **Response to Reviewer KHEq**
>
> We thank the reviewer for the helpful and constructive comments.
>
> $\\textbf{Q1}$: The experiment settings lack some explanations, e.g. the reason for the specific value of hyper-parameters.
>
> $\\textbf{A1}$: There are mainly two types of hyper-parameters in our work.
>
> (1) Hyper-parameters of the model architecture (such as the depth of transformer and the learning rate). For these hyper-parameters, we actually use grid-search to determine the best choices of them in our experiments. We will state more clearly in the paper.
>
> (2) The batch size of state-action pairs. During training, we feed a batch of state-action pairs to the Transformer in order to compute the approximated policy representations. To balance the computational cost and the performance, we chose the number 256 as the batch size. This choice is supported by the experimental results bellow, which show the averaged rank correlations of our model with the batch size growing. We can find that when the batch size is larger than 256, the performance of our model tends to be stable.
>
> $$\\begin{array}{|c|c|c|c|c|c|c|c|c|c|}
> \\hline\text{Batch Size} &8	&16	&32	&64&	128	&256&	512&	1024&	2048\\\\
> \\hline \text{Avg. Rank Correlation} &0.21&	0.27&	0.37&	0.39&	0.56&	0.65	&0.64&	0.66	&0.65\\\\
> \\hline
> \\end{array}$$
>
> $\\textbf{Q2}$: Some other label aggregation methods should also be added.
>
> $\\textbf{A2}$: We agree with the reviewer that other label aggregation methods should be compared. We add two simple label aggregation methods, namely "Average Score" and "Majority Voting", and compare their rank correlations with the Crowd Layer (CL). We can see from the following table that our method domindates these two baselines in all of the six environements. However, we agree that CL may not be the best method for learning from crowd, since this line of research is evolving itself. In fact, our contribution is approaching the off-policy ranking problem from a crowdsourcing perspective, therefore any advances in crowdsourcing or label aggregation could be combined with our proposed SOCCER framework and benefit the OPR task.
>
> $$\\begin{array}{|c|c|c|c|}
> \\hline\text{Environment} &\text{Avg. Score}&\text{Major Voting}&\text{Ours}\\\\
> \\hline HalfCheetah-expert &-0.34	&-0.27	&\textbf{0.71}\\\\
> \\hline HalfCheetah-full-replay &0.24&	0.31&	\textbf{0.74}\\\\
> \\hline HalfCheetah-medium &0.32&	0.57&	\textbf{0.81}\\\\
> \\hline Walker2d-expert &0.53	&0.23&	\textbf{0.85}\\\\
> \\hline Walker2d-full-replay &0.41&	0.31&	\textbf{0.82}\\\\
> \\hline Walker2d-medium &0.21&	0.29&	\textbf{0.80}	\\\\
> \\hline
> \\end{array}$$
>
> $\\textbf{Q3}$: The ablation experiment introduces an extra strategy to obtain ‘truth’ to train the method. This setting lowers the convincing performance of the proposed method.
>
> $\\textbf{A3}$: It seems that the reviewer has some misunderstandings to our design of the ablation study. We will state more clearly in the next version of our paper. In fact, by comparing SOCCER with SOCCER-without-CL (trained using ground-truth labels), we demonstrate the superiority of SOCCER since it can achieve comparable performance with SOCCER-without-CL, but does not require the ground-truth labels. Moreover, to demonstrate the effectiveness of our proposed Policy Comparison Transformer, we train SOCCER-without-CL and SOPR-T using the same set of ground-truth labels for a fair comparison. Therefore, introducing ground-truth labels are actually necessary in our ablation studies.

---

### Official Review · Reviewer_jfpk · 2022-10-25

**Confidence:** 4
**Correctness:** 3
**Technical Novelty And Significance:** 3
**Empirical Novelty And Significance:** 3
**Recommendation:** 6

**Clarity, Quality, Novelty And Reproducibility:**

Clarity: good.

Quality: good.

Novelty: the paper combines several existing components and aggregates the results of the existing methods, but the idea of the aggregation is reasonably novel. Also, using pairwise instead of direct policy representation sounds novel to me.

Reproducibility: the paper provides sufficient details on the methodology as the space permits. Are the authors planning to open source the code?

**Strength And Weaknesses:**

Strengths:

- The paper is well written and easy to follow. The details of the method and the experiments are clearly explained. The figures are informative and well explained.
- The idea of using pairwise policy representation sounds interesting and novel.
- The experimental results where the proposed method outperformed the other baseline is very encouraging.
- The experimental results studies the problem with different settings.

Weaknesses:

- To my mind, the experimental results are lacking an adequate baseline that is comparable with the proposed method. A baseline would be comparable if it also uses other existing OPE methods to aggregate the results. For example, I can imagine several easy baselines in this case: 1) take the average OPE scores of all methods and produce a ranking out of them, 2) use a majority voting scheme to aggregate the rankings, 3) there are many rank aggregation methods that could be considered, for example, [1].
- As the proposed method aggregates the ranks from the existing policies, the computational cost for it is much higher than for any other method and it includes the costs of all other methods. This should also be discussed.
- Another limitation in the current experiments is that as far as I understand the method needs to be trained for every new set of policies and the environment from scratch. Then, it is tested on its own training set (no validation or test set). Do I understand the setting correctly? Would the policy representations generalize across different sets of policies? Suppose a new policy is added to the set of policies, can the previous results be re-used? To me, the method would be useful in practice if it can show signs of such generalization.
- I do not understand why transformers are the best architecture in the given policy representation design. As the states (equation 1) are chosen as just a set (not ordered), what is the advantage of using a transformer which is known to be the best suited for sequential data? Did the authors consider other architectures (possibly simpler, e.g., MLP) here?

Other comments:
- Several times the authors mention that in practice finding the best policy is the main objective. In that case, it would be more logical to consider off-policy policy selection (OPS) problem formulation and as the quality metric measure the regret @1. How would the method perform in that case?
- I still do not understand the role and training of the "aggregation token" very well, maybe this could be explained further.
- In the first part of section 4.2 the authors say that they "show how the policy ranking can be reduced to binary classification". I think this is a common way to approach the ranking problem (but the text sounds now like this is one of the contributions). Some work could be references here, for example, [2].

[1] Fast and Accurate Inference of Plackett–Luce Models. Lucas Maystre, Matthias Grossglauser. NIPS 2015.

[2] Preference Learning with Gaussian Processes. Wei Chu, Zoubin Ghahramani. ICML 2005.


**Summary Of The Paper:**

This paper proposes a new method for ranking of offline RL policies with off-policy evaluation (OPE). The ranking is produced with a model that 1) learns a pairwise policy representation with a transformer architecture, 2) uses a crowd layer to aggregate OPE scores of other methods. In the experimental results the authors show that their method is able to outperform the other baselines. The ablation studies show the importance of various components of the proposed method.

**Summary Of The Review:**

I am leaning toward rejecting this paper mainly because I find the experiments lacking comparable baselines that would benefit from aggregating the results of the existing methods in the same way as the proposed method. Also, I would also like to see some generalization of the method or policy representation to the unseen policies that would make the method scalable to real world problems.

---
Updated my score after rebuttal in the light of new empirical results.

---

> ### Author Response · Authors · 2022-11-19
> **Responses to Reviewer jfpk (Part 2)**
>
> $\\textbf{Q4}$: Why transformers are the best architectures in the given policy representation design.
>
> $\\textbf{A4}$: Transformers are powerful due to its self-attention mechanism which can capture the correlations between input tokens. Moreover, the design of positional encoding makes Transformers work well on sequential data. In our Policy Comparision Transformer (PCT), the self-attention machanism successfully captures the correlation between state-action pairs, which is key to learn effective and generalizable policy representations. Since we want to learn pairwise policy representations, we design a novel positional encoding which reflects the $\\textbf{order of two policies}$, so that the learned pairwise policy representations can be directly used to predict the order of them. To show the advantage of Transformers, we did additional ablation experiments by replacing the PCT with an MLP which has the same number of parameters. We present the average rank correlation of PCT and MLP of the 6-policy set in HalfCheetah-v2 environment as follows. We can see that PCT indeed performs much better MLP.
> $$\\begin{array}{|c|c|}
> \\hline\text{ } &\text{Rank Correlation}\\\\
> \\hline \text{PCT} &0.65\\\\
> \\hline \text{MLP} &0.32\\\\
> \\hline
> \\end{array}$$
>
> $\\textbf{Q5}$: The role and training of the "aggregation token”.
>
> $\\textbf{A5}$: The aggregation token is similar to the "cls" token which is widely used in other transformer architectures, e.g. ViT and BERT. In our case, this token aggregates information from state-action pairs and will be used in predicting the order of two policies.

---

> > ### Comment · Reviewer_jfpk · 2022-11-29
> > **Thank you for providing the additional results**
> >
> > I would like to thank the authors for providing the additional results supporting the paper's claims.
> >
> > - I find it informative that average score and majority voting schemes do not work well in this setting, and that using architecture without attention mechanism results in the large drop in performance.
> > - The generalisation results are quite interesting and might require further discussion. Why in some cases it is better to train on a different set of policies? In which cases training on a different set of policies does not work very well? It might be a useful finding for practical considerations.
> > - Regarding the computational costs, I still think that the authors should still discuss this in the paper. Even through the method relies on existing OPE techniques, it still means that for any particular problem, the practitioner needs to train *all* the OPE methods instead of just one of them before following the proposed methodology.
> > - Related to this, other reviewers raised concerns about the hyperparameters of the method and I agree that this should be clearly explained.
> >
> > In the light of new empirical results, I update my score as my biggest concerns were addressed.

---

> > > ### Author Response · Authors · 2022-12-10
> > > **Thanks for your insightful suggestions.**
> > >
> > > 1. We agree that there would be more interesting findings from more extensive experiments regarding to the generalisation ability.
> > >
> > > 2. Yes we will discuss about the computational cost and the choice of hyperparameters in the next version of our paper. Thank you for your advise!

---

> ### Author Response · Authors · 2022-11-19
> **Responses to Reviewer jfpk (Part 1)**
>
> We thank reviewer for your helpful comments and kind suggestions.
>
> $\\textbf{Q1}$: The experimental results are lacking an adequate baseline that is comparable with the proposed method.
>
> $\\textbf{A1}$: The reviewer suggests to compare the Crowd Layer (CL) with other label aggregation methods such as "Average Score" and "Majority Voting". Actually, the superiority of CL over these baslines has been demonstrated in [1]. However, we agree with the reviewer that it is still valuable to reproduce this superiority in the context of off-policy ranking. Specifically, we use the methods of "Average" and "Majority Voting"  to replace the CL in SOCCER and report the rank correlations in the following table. We can see from the first four columns that our method domindates these two baselines in all of the six environements. Note that the framework of SOCCER can also combine with more advanced crowdsourcing methods other than CL.
>
> We note that the reviewer mentioned Rank Aggregation (RA), which is a line of works that aggregate a set of pairwise comparisons into a ranking list. Since RA aggregates pairwise comparisons instead of labels,  we cannot directly compare it with CL. In fact, SOCCER also incorperates a simple averaged ranking aggregation procedure, as described in Equation (6) in the paper. Following the reviewer's suggestion, we use a more recent and simple RA method [2] to replace Equation (6). From the last two columns we can see that the method in [2] indeed further improves SOCCER in some of the test evironments. However, this does not contradicts to our main contribution: modeling the off-policy ranking problem from the perspective of crowdsourcing.
> $$
> score_i = \frac{1}{N}\sum_{j\\neq i}\hat{y}_{i,j}, \\,\\, i=1,...,N \\,\\,\\,\\,\\,\\,\\,\\,\\,\\,(6)
> $$
>
> $$\\begin{array}{|c|c|c|c|}
> \\hline\text{Environment} &\text{Avg. Score}&\text{Major Voting}&\text{Ours}&\text{Ours with RA}\\\\
> \\hline HalfCheetah-expert &-0.34	&-0.27	&0.71&	\textbf{0.72}\\\\
> \\hline HalfCheetah-full-replay &0.24&	0.31&	\textbf{0.74}&	0.73\\\\
> \\hline HalfCheetah-medium &0.32&	0.57&	\textbf{0.81}&	0.80\\\\
> \\hline Walker2d-expert &0.53	&0.23&	\textbf{0.85}&	0.83\\\\
> \\hline Walker2d-full-replay &0.41&	0.31&	\textbf{0.82}&	0.75\\\\
> \\hline Walker2d-medium &0.21&	0.29&	0.80	&\textbf{0.87}\\\\
> \\hline
> \\end{array}$$
>
> $\\textbf{Q2}$: As the proposed method aggregates the ranks from the existing policies, the computational cost for it is much higher than for any other method and it includes the costs of all other methods. It should be discussed.
>
> $\\textbf{A2}$: Since our method is built upon existing OPE methods, we do need to implement these methods. However, we don't think it is a big problem due to three facts. (1) Existing OPE methods can be implemented in parallel. (2) The computational cost of our proposed Policy Comparison Transformer and Crowd Layer depends on the amount of training data, and it is usually worth to incorporate more data if they are available. (3) Our model, once trained, can be used to inference the rankings of arbitray number of policies. By contrast, existing OPE methods such as FQE and model-based approaches, need to train their models from scratch for every new policy. Therefore, from this perspective, the averaged computational cost of our method is much lower than existing methods.
>
> $\\textbf{Q3}$: The generalization ability of our model.
>
> $\\textbf{A3}$: We thank the reviewer for pointing out this issue. The generalization ability is actually an advantage of our method, due to the strong representational ability of our proposed Policy Comparison Transformer. To test the generalization ability, we added a new set of experiments where our model is trained on a given policy set and tested on other sets. We report the rank correlations in the following table. Set 1-5 represent Halfcheetah-expert set I, Halfcheetah-expert set II, Halfcheetah-full-replay set I, Halfcheetah-full-replay set II, Halfcheetah-medium set I, respectively. Specifically, the rows indiate the policy sets we used for training and the columns indicate the policy sets we used for test. We can see that our method generalizes well in most cases. For example, the model trained using Set 1 achieves 0.52 correlation when tested on Set 2 (see row 1, colunm 2), which even outperforms the model trained using Set 2 itself.
>
> $$\\begin{array}{|c|c|c|c|c|}
> \\hline\text{ } &\text{Set 1}&\text{Set 2}&\text{Set 3}&\text{Set 4}&\text{Set 5}\\\\
> \\hline \text{Set 1} &\textbf{0.71}	&\textbf{0.52}&	0.65&	0.32&	0.65\\\\
> \\hline \text{Set 2} &0.51&	0.42&	0.52	&0.53&	0.76\\\\
> \\hline \text{Set 3} &0.66	&0.43&	\textbf{0.74}&	\textbf{0.65}&	0.77\\\\
> \\hline \text{Set 4} &0.43	&0.42	&0.54&	0.41	&0.50\\\\
> \\hline \text{Set 5} &0.65	&0.32&	0.67	&0.40&	\textbf{0.81}\\\\
> \\hline
> \\end{array}$$
>
> [1] Deep learning from crowds. In AAAI. 2018.
>
> [2] Just sort it! A simple and effective approach to active preference learning. In ICML. 2017.

---

### Decision · Program_Chairs · 2023-01-20

**Decision:**

Reject

**Justification For Why Not Higher Score:**

This work may not be sound.

**Justification For Why Not Lower Score:**

N/A

**Metareview: Summary, Strengths And Weaknesses:**

This paper builds on a 2022 work where a policy comparator is trained in a supervised learning fashion on pairs of policies. This work makes two additional advancements:

* A better representation for pairs of evaluated policies.

* A crowdsourcing framework where the performance of the policies is estimated using existing off-policy estimators. The logic behind this approach is that if most off-policy estimators are pretty good, it is possible to pool them to learn even a better comparator.

This paper received 3 borderline reviews and one reject. The borderline reviews slightly improved after the rebuttal but the reject review remained. The main reason for this review is that the approach is not principled, because it is a result of self-training from potentially biased and correlated off-policy estimates. At the last minute (December 9), the authors submitted a formal analysis. The most critical reviewer checked it and found strong assumptions, such as independence of off-policy estimators. This is obviously false because the estimators use the same dataset.

To break the impasse between the authors and the reviewer, I read the paper and also the new analysis. In summary, Theorem 1 is proved for

* Any two fixed policies $\pi_i$ and $\pi_j$

* A fixed OPE worker

The random quantities are the biases in the OPE of $\pi_i$ and $\pi_j$. The proof of Theorem 1 relies on an assumption that the biases are independently distributed. However, how can this be if $\pi_i$ and $\pi_j$ are very similar? One approach, as the most critical reviewer pointed out, is that the OPE of $\pi_i$ uses a different dataset from that of $\pi_j$. This is not done in the paper. Interestingly, this is also the main issue that the most critical reviewer had since the beginning. Everything is done on a single dataset and it is unclear how the biases average out.

After reading the paper, I also wanted to share one observation. If you check Figure 4, SOCCER without CL seems to be better overall than SOCCER. Thus the crowdsourcing idea may not work. Therefore, I believe that the main lift in this paper is due to a new way of learning the representation for pairs of policies.